# Critically Appraised Topic on Canine Leishmaniosis: Does Treatment with Antimonials and Allopurinol Have the Same Clinical and Clinicopathological Efficacy as Treatment with Miltefosine and Allopurinol, after One Month of Treatment?

**DOI:** 10.3390/vetsci11060231

**Published:** 2024-05-22

**Authors:** Marina García, Lluís Ferrer, Laura Ordeix

**Affiliations:** Departament de Medicina i Cirurgia Animals, Fundació Hospital Clínic Veterinari, Universitat Autònoma de Barcelona, 08193 Bellaterra, Spain; marina.garcia.acedos@uab.cat (M.G.); lluis.ferrer@uab.cat (L.F.)

**Keywords:** dogs, leishmaniosis, miltefosine, meglumine antimoniate, allopurinol

## Abstract

**Simple Summary:**

To treat canine leishmaniosis, two medication combinations are used: meglumine antimoniate with allopurinol, or miltefosine with allopurinol. This study compared the clinical effectiveness of both options after one month of treatment. The results showed a significantly higher rate of complete clinical cure in the meglumine antimoniate and allopurinol group. Most dogs in both groups improved after one month of treatment. There were no significant differences in the presence of the *Leishmania* parasite between the groups after treatment. The humoral immune response could not be compared due to insufficient data. Overall, meglumine antimoniate and allopurinol seem more effective in treating canine leishmaniosis compared to miltefosine and allopurinol.

**Abstract:**

The treatment of canine leishmaniosis commonly involves meglumine antimoniate with allopurinol or miltefosine with allopurinol. This study aims to compare the clinical and clinicopathological efficacy at 28–30 days of conventional dosing regimens for both treatments using the critically appraised topic methodology. A comprehensive search across three databases (PubMed, CAB Abstracts, and Web of Science) from March 2004 to September 2023 yielded 16 relevant articles, encompassing 325 ogs treated with meglumine antimoniate and allopurinol, and 273 dogs treated with miltefosine and allopurinol. The findings indicated a significantly higher rate of complete clinical cure in the group treated with meglumine antimoniate and allopurinol. Most dogs in both groups exhibited improvement in clinicopathological alterations after one month of treatment. No significant difference was observed in the number of dogs that showed a negative *Leishmania* qPCR between the two groups, one month post-treatment. However, quantitative serology results were not commonly reported in the available data and therefore this aspect could not be compared.

## 1. Introduction

Canine leishmaniosis (CanL) caused by *Leishmania infantum* is a severe and highly prevalent disease in the Mediterranean basin. Despite widespread preventive measures, such as the use of insecticidal repellents and vaccines, many dogs develop clinical diseases and require treatment. Standard treatment for CanL involves the use of pentavalent antimonials or miltefosine both in combination with allopurinol [1,2]. Although less explored, secondary drugs such as paromomycin and marbofloxacin have also been shown to have antileishmanial effects [3]. The standard treatment protocol in Europe for dogs in stable clinical condition is allopurinol at 10 mg/kg every 12 h orally, in combination with subcutaneous injections of meglumine antimoniate at 100 mg/kg every 24 h for 28 days, or in combination with miltefosine at 2 mg/kg orally every 24 h for 28 days [2,3]. Meglumine antimoniate is the most used parasiticidal agent in CanL. It inhibits the parasite’s metabolism by blocking the enzyme phosphofructokinase. Additionally, it enhances the phagocytic activity of macrophages [3,4]. The meglumine antimoniate is commonly used in conjunction with allopurinol during the first four weeks of treatment. Miltefosine is an alkylphosphocholine that inhibits the biosynthesis of the glycosylphosphatidylinositol receptor, which is crucial for the intracellular survival of *Leishmania*. It interferes with the synthesis of phospholipase and protein kinase B, which can affect the biosynthesis of glycolipids and membrane glycoproteins of the parasite, leading to apoptosis [5,6]. Additionally, studies suggest that miltefosine has immunomodulatory properties [7]. Miltefosine can be orally administered during the initial month of treatment in conjunction with allopurinol. Finally, allopurinol is the main drug used for the treatment of CanL. It is a hypoxanthine analog, alters purine metabolism, and inhibits *Leishmania* growth by blocking xanthine oxidase [8]. The combination of two drugs, one parasiticidal and one with a more parasitostatic effect, reduces the frequency of relapses and probably also reduces the development of resistance. The use of meglumine antimoniate or miltefosine as monotherapy has not been demonstrated to provide consistent long-term suppression of parasite loads in dogs and may lead to disease relapse [9].

The clinical efficacy of both protocols, pentavalent antimonials or miltefosine, both in combination with allopurinol, has been demonstrated to treat CanL [1]. The most usual protocol involves administering meglumine antimoniate or miltefosine for one month and allopurinol for a minimum of 6 months. After one month of treatment, the first follow-up visit is carried out to assess the clinical improvement of the patients. This first post-treatment visit is of critical importance to decide whether the patient has responded well to treatment or whether the parasiticide treatment needs to be extended or changed [1].

This study aims to compare the efficacy of both treatments in the control of clinical and clinicopathological alterations associated with leishmaniosis after 28–30 days of treatment using the critically appraised topic (CAT) methodology.

## 2. Materials and Methods

Clinical scenario: A 13-year-old female sterilized German shepherd was referred for consultation due to nasal ulcers. On physical examination, mild bilateral submandibular and popliteal lymphadenomegaly was observed. A dermatological examination revealed two ulcers covered with crusts in the nasal folds. Differential diagnoses included leishmaniosis, discoid lupus erythematosus, or mucocutaneous pyoderma. Impression smears revealed a non-specific neutrophilic inflammation, without the presence of bacteria. A complete blood count (Sysmex XN-1000V, Symex Corporation, Nordestedt, Germany) and complete biochemical profile (Olympus AU400; Olympus Diagnostica GmbH, Hamburg, Germany) revealed mild lymphopenia and hyperproteinemia. The serum protein electrophoresis (Minicap, Sebia, Evry, France) showed an increase in gamma globulins, the urinary protein creatinine ratio (UPC) was <0.5 (non-proteinuric), and an ELISA for anti-*Leishmania infantum* antibodies yielded a highly positive result of 375 EU (≥300 EU). Considering together the clinical signs, the clinicopathologic changes, and the high level of anti-*Leishmania* antibodies, CanL was diagnosed. However, it was unclear whether the ulcerative nasal lesions were a consequence of *Leishmania* infection or had a different etiology. The most effective way to elucidate the cause of nasal lesions is to perform a histopathological study of the lesions, with immunohistochemical or molecular tests to detect the presence of the parasite in the lesions. A less invasive alternative is to assess whether the lesions resolve after treatment for leishmaniosis. If this is the case, the lesions are considered to be a consequence of the infection and if not, the lesions are histopathologically studied. After discussing the two alternatives with the dog’s guardians, it was decided to proceed with treatment with meglumine antimoniate and allopurinol, for both therapeutic purposes and to clarify the diagnosis. 

However, the pet guardian expressed concern about the difficulty of administering the daily subcutaneous injections and inquired about the possibility of using oral medication instead. Therefore, it was proposed to replace the meglumine antimoniate with miltefosine, which is administered by mouth. The pet guardian, however, wanted to know if both strategies had the same efficacy.

Structured question: Does treatment with miltefosine and allopurinol have the same clinical and clinicopathological efficacy as treatment with antimonials and allopurinol in dogs with leishmaniosis one month after therapy?

Search strategy: The CAB Abstracts, PUBMED, and Web of Science databases were examined using the following search strategy: (dog OR dogs OR canine OR canines) AND (miltefosine OR meglumine antimoniate OR antimoniate OR meglumine) AND (leishman*) AND (allopurinol) AND (clinical OR clinicopathological). The search was restricted from 12 March 2004, when a systematic review of the treatment of leishmaniosis was published [10], to September 2023. The selection process focused on clinical studies, excluding narrative reviews, book chapters, and congress proceedings. Moreover, in non-English written studies and those in which meglumine antimoniate or miltefosine was used without allopurinol, non-standardized doses were used, and those articles that did not present information 28–30 days after starting treatment were excluded from the analysis.

Critical analysis: To assess clinical efficacy, the percentage reduction in dogs’ clinical scores between day 0 and day 30 of treatment was calculated in any study. Individuals who achieved a clinical score reduction of more than 90% were classified as having complete remission (CR), while those with a clinical score reduction of less than 90% were classified as having partial remission (PR). The number of dogs with CR were compared among both groups of treatment. Clinical-pathologic efficacy was evaluated by calculating the mean percentage reduction in total serum proteins and globulins, and the mean percentage increase in hematocrit and albumin. When possible, reduction in parasite load and antibody level was also assessed, calculating the percentage reduction of qPCR performed in lymph nodes, blood, bone marrow, or skin and of indirect fluorescent antibody tests (IFATs) or ELISA, respectively. The number of dogs with positive or negative qPCR in any tissue or fluid was compared among both groups of treatment. The statistical study was categorical and analyzed using the Fisher test. *p* < 0.05 was considered the critical level of significance (Statistical calculators online, https://www.socscistatistics.com/tests/ (accessed on 18 March 2024) Social Science Statistics).

## 3. Results

The literature search identified 81, 85, and 102 citations in the PUBMED, CAB Abstracts, and Web of Science databases, respectively. Finally, a total of 16 articles were selected in accordance with the search strategy criteria stated above.

The selected 16 articles, which were subjected to a thorough evaluation, comprised a systematic review [10], one randomized controlled trial [11], two cohort studies [12,13], two case/control studies [7,14], five case series [6,15,16,17,18], and five case reports [19,20,21,22,23]. The quality of the evidence is shown in Figure 1. Table 1 presents the relevant data extracted from each of the 16 selected articles and the analysis of the different studies. In total, these studies included the evaluation of 598 dogs diagnosed with leishmaniosis; 15 dogs were subtracted from the total as they were shared in two publications [12,13].

### 3.1. Meglumine Antimoniate plus Allopurinol

Eight articles were identified describing the use of this therapeutical combination, of which one was a systematic review containing nine studies of interest [24,25,26,27,28,29,30,31,32]. Three hundred twenty-five dogs reported in these 16 articles were reviewed.

The clinical efficacy after one month of treatment was mentioned only in eight out of the sixteen articles [7,11,14,15,16,18,21,28,29,31,32]. A total of 148 dogs out of 325 dogs could be evaluated for clinical efficacy. A total of 14 out of 148 dogs (9.5%) demonstrated CR, meanwhile, 134 out of 148 dogs (90.6%) showed PR 30 days after the start of treatment.

Hematocrit values were evaluated in two out of sixteen articles describing the changes observed in 43 out of 325 dogs [7,18]. A mean increase of 8.6% and 8.3% of the hematocrit was observed in six and thirty-seven dogs, respectively [7,18]. Total protein levels were examined in three out of sixteen articles studying a total of 96 out of 325 dogs. A decrease of 2.5%, 9%, and 15% was observed in all these dogs [7,16,18]. Albumin levels were evaluated in two out of sixteen studies, involving a total of 59 dogs. An increase of 16% and 20% in albumin levels was described in six and fifty-three dogs, respectively [16,18]. Globulin levels were assessed in three out of sixteen articles evaluating 96 dogs. A decrease of 24%, 33.6%, and 44% was described in fifty-four, six, and thirty-seven dogs, respectively [7,16,18].

One month after treatment parasite DNA could be detected in lymph nodes and blood [10,11,14,18]. *Leishmania* qPCR from lymph nodes was analyzed in three articles, involving a total of 30 out of 325 dogs. A mean decrease of 18%, 28.6%, and 80% in the parasite load was described in nine, seven, and fourteen dogs, respectively [10,14]. The mean reduction of the blood qPCR in 37 dogs was 11.27% [18]. Bone marrow qPCR was analyzed in only one article and was negative in all 37 dogs studied [11].

*Leishmania* serology performed one month after treatment was evaluated only in two out of sixteen studies, using indirect fluorescent antibody tests (IFAT) or ELISA in nine and thirty-seven dogs, respectively. Following treatment, the first study described a 53% decrease [14], while the second study observed a 44% decrease [18].

### 3.2. Miltefosine plus Allopurinol

Ten articles were reviewed in which this therapeutical combination was used, with a total of 273 dogs evaluated [6,7,11,12,13,17,19,20,22,23].

The clinical efficacy could only be assessed in 267 dogs. All reported dogs showed partial improvement within 30 days of starting treatment [6,11,13,20].

Hematocrit was assessed only in two out of ten studies, involving a total of 21 dogs out of 273. In six dogs, a decrease of 6.3% was observed [7], while in the other study evaluating fifteen dogs an average increase of 18% was reported [12]. Total proteins were evaluated in three out of the ten studies, involving a total of 23 dogs out of 273. In two studies, a mean decrease of 11.2% and 19.6% was noted [7,21]. One study reported a mean increase of 0.28% in 15 dogs [12]. Albumin levels were evaluated in two out of ten studies, with a total of eight dogs out of two hundred and seventy-three dogs described. An average increase of 6.34% and 32% in the albumin concentration were described [7,20]. The value of globulins could be evaluated in two out of the ten studies, involving a total of 8 dogs out of the 273 dogs. In one study, with a total of two dogs, a decrease of 13.5% was observed [20]. In the other study, with a total of six dogs, an increase of 2.7% was observed [7].

Although in different samples, parasite DNA was studied by means of qPCR in five studies out of ten. Lymph node *Leishmania* qPCR was analyzed in two studies, involving a total of 37 dogs out of 273 dogs. All dogs were still positive one month after initiation of therapy. However, a reduction of 67% and 87% of the parasite load was observed [6,14]. Blood *Leishmania* qPCR was studied in one study involving a total of 37 dogs and a mean reduction of 29.7% of the parasite load was observed [6]. Skin *Leishmania* qPCR was analyzed in one article involving a total of 15 dogs, in which a mean reduction of 95% of the parasite load was detected (13). Bone marrow *Leishmania* qPCR was negative after one month of initiating therapy in 36 dogs described in one study [11].

Information regarding *Leishmania* serology after one month of therapy was not reported in any of the studies.

### 3.3. Compared Aspects between the Two Groups

Comparison of the articles was difficult due to differences in study designs, the number of dogs analyzed for each parameter under study, and type of tests and samples assessed in the studies. However, some aspects could be compared.

Complete clinical remission was observed in a major proportion of dogs treated with meglumine antimoniate plus allopurinol than in dogs treated with miltefosine and allopurinol (*p* < 0.00001, Fisher exact test).

Concerning the qPCR results, the comparison involved assessing test positivity or negativity across all organs collectively. Notably, there was no significant difference in the number of negative samples at one month of treatment between the groups (*p* = 0.32, Fisher exact test).

## 4. Discussion

The medical treatment for CanL is tailored to the specific clinical stage of the infected dog (9). Although the duration of treatment is typically extended, usually dogs are initially treated with meglumine antimoniate in combination with allopurinol for the first 4 weeks of treatment. Alternatively, miltefosine, can be used orally for the first month of treatment in combination with allopurinol instead of meglumine antimoniate [9]. In previous studies, miltefosine has shown good tolerability, with the exception of mild, self-limited gastrointestinal side effects [17].

The main objectives of treatment are the resolution of the clinical picture and the normalization of clinicopathological values. Although it is known that the treatment does not lead to a complete elimination of the infection (sterility), it is expected to also cause a marked reduction in the parasite load. Effective treatment should result in the resolution of clinical signs; however, the time in which this happens is not fully defined by either the type of disease or the type of treatment used.

In seropositive dogs, in the absence of a direct correlation between the clinical signs and the presence of *Leishmania* (by parasitological direct tests), the response to treatment is often used as the definitive method to attribute a role to the infection in the clinical manifestations. Erosive-ulcerative nasal dermatitis as the one presented in the clinical scenario is an example of this situation. It is a clinical and pathological manifestation suggestive of leishmaniosis as well as of discoid lupus erythematosus [9,33]. Therefore, early assessment of the response or the lack of response to anti-*Leishmania* treatment is of great importance in the clinical decision-making process in some cases. Hence, the objective of this CAT and comparative analysis was to enhance our understanding of the potential distinctions among conventional leishmaniosis treatment strategies. The focus was on evaluating their clinical and clinicopathological efficacy, along with assessing the impact on parasite load and antibody levels after one month of treatment.

Comparing the selected articles proved challenging owing mainly to variations in study designs and the varying number of dogs analyzed for each parameter. Notwithstanding these challenges, it appeared that the meglumine antimoniate combined with allopurinol exhibited greater clinical effectiveness after one month of treatment compared to the miltefosine and allopurinol combination, due to a significantly higher proportion of dogs with complete clinical remission within the initial month of treatment with meglumine antimoniate with allopurinol.

The prevalent clinical-pathological manifestations observed in CanL encompass mild to moderate normocytic normochromic and non-regenerative anemia, lymphopenia, hyperproteinemia characterized by elevated gamma-globulins and beta-globulins, along with hypoalbuminemia, leading to a reduced albumin/globulin ratio and proteinuria [9,34]. While the comparison of the two therapeutic strategies proved challenging due to the limited analysis of most data, most dogs tested in both groups exhibited improvement in their clinic-pathological alterations within the initial month of treatment. An exception was noted in only one study within the miltefosine and allopurinol group, where dogs showed an decrease in hematocrit and increase gamma globulins [7]. The possible causes of this unexpected finding were not discussed by the authors of the article.

A reduction in parasite load was observed in the blood, lymph nodes, bone marrow, and skin during the initial 30-day period of both treatments. Furthermore, when evaluating test positivity or negativity across all tissues or fluids collectively, there was no significant difference in the number of negative samples at one month of treatment between the groups. Hence, it appears that the performance of this test may not serve as a differentiating strategy for assessing the response to treatment, at least during the initial 30-day period.

Information regarding the outcomes of serological tests, such as ELISA and IFAT, conducted 30 days post-treatment, was notably lacking in the majority of studies analyzed herein. This absence could be attributed to the less common practice of performing these tests until at least 3 to 6 months after the beginning of treatment [9]. It is suggested that dogs, particularly those with high antibody levels at diagnosis, may not exhibit a discernible reduction in antibody levels within the initial 30 days in commercial assays (i.e., no end point sera dilution tests) [18]. Typically, a substantial decrease in levels becomes noticeable around 6 months, primarily influenced by the extended half-life of IgG and the type of test performed [2,18,35,36].

## 5. Conclusions

Based on the evidence reviewed, it can be concluded that after one month of treatment, the combination of meglumine antimoniate and allopurinol shows better clinical efficacy in the treatment of canine leishmaniosis than the combination of miltefosine and allopurinol. This is particularly relevant in cases of *Leishmania* infection where treatment is carried out for confirmatory diagnostic purposes (*ex juvantibus* diagnosis).

## Figures and Tables

**Figure 1 vetsci-11-00231-f001:**
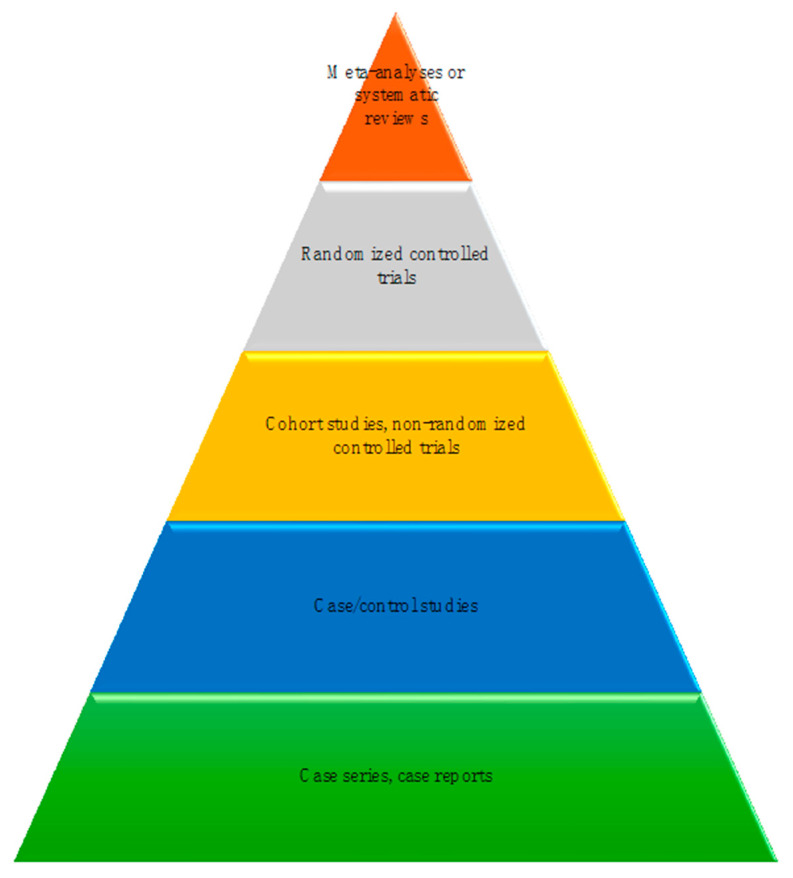
Strength of evidence.Meta-analyses or systematic reviews [10]; Randomized controlled trials [11]; Cohort studies, non-randomized controlled trials [12,13]; Case/control studies [7,14]; Case series, case reports [6,15,16,17,18,19,20,21,22,23].

**Table 1 vetsci-11-00231-t001:** Quality of the evidence.

Reference	Study Design	N	Treatment	Results	Limits
**Noli C. et al. (2005)** [10]	Systematic review	N = 873 dogs N (AM + A) = 160 dogs	AM: 40–100 mg/kg/SID SC (3–6 weeks) A: 10–30 mg/kg/BID PO (4 weeks–9 months)	Complete and partial clinical remission in 96–100% of dogs treated with AM doses at 100 mg/kg/24 h SC for 3–4 weeks.	No clinical-pathological efficacy data (28–30 days). No parasitological data (28–30 days).
**Manna L. et al. (2009)** [6]	Case series	N = 28 dogs N = 15 healthy	M: 2 mg/kg/SID PO (30 days) A: 10 mg/kg/BID PO (12 months)	Partial clinical remission (74%). Parasitic load remission lymphonodes PCR in 30 days (87%). Parasitic load remission in blood PCR in 30 days (29.7%).	Study design.
**Manzillo V.F. et al. (2009)** [19]	Case report	N = 1 dog	M: 2 mg/kg/SID PO (4 weeks) A: 10 mg/kg/BID PO (6 months)	Partial clinical remission in 28 days. Partial clinical-pathological in 28 days.	Study design. No parasitological data (28–30 days).
**Miro G. et al. (2009)** [11]	Randomized controlled	N (G1) = 36 dogs N (G2) = 37 dogs	G1: M: 2 mg/kg/SID PO (28 days) A 10 mg/kg/BID PO (7 months) G2: AM: 50 mg/kg/BID SC (28 days) A: 10 mg/kg/BID PO (7 months)	Partial clinical remission in 28 days (G1 46.3% and G2 43.2%). Parasite load remission in bone marrow PCR in 29 days (G1 100% G2 100%).	No clinical-pathological efficacy data (28–30 days). Non-independent clinical research.
**Torres M. et al. (2011)** [15]	Case series	N = 23 dogs	AM: 100 mg/kg/SID SC (28 days) A: 10–30 mg/kg/BID PO (at least a month)	Partial clinical remission in 28 days (43%).	Study design. No parasitological data (28–30 days).
**Pierantozzi M. et al. (2013)** [16]	Case series	N = 53 dogs	AM:75–100 mg/kg/SID or BID SC (28 days) A: 10–20 mg/kg/BID or SID PO (4–8 weeks)	Clinical-pathological in 28 days: Total protein decrease (15%). Albumin increase (20%). Globulin decrease (24%).	Study design. No parasitological data (28–30 days).
**Proverbio D. et al. (2014)** [20]	Case report	N = 2	Case 1: M: 2 mg/kg/SID PO (28 days) A: 10 mg/kg/BID PO (390 days) Case 2: M: 2 mg/kg/SID PO (28 days) A: 10 mg/kg/BID PO (28 days)	Case 1: Partial clinical remission in 28 days (63%). Clinical-pathological in 28 days: Total protein decrease (16%). Albumin increase (11%). Globulin decrease (12%). Case 2: Partial clinical remission in 28 days (36%). Clinical-pathological remission in 28 days: Total protein decrease (6,4%). Albumin increase (1,69%). Globulin decrease (15%).	Study design.
**Manna L. et al. (2015)** [14]	Case/Control Study	N (G1) = 9/18 N (G2) = 9/18	G1: AM: 100 mg/kg/SID SC (30 days) A: 10 mg/kg/BID PO (6 years) G2: M: 2 mg/Kg/SID (30 days) A: 10 mg/kg/BID (6 years)	Partial clinical remission in 28 days (G1 71.1% and G2 71.7%). Parasitic remission lymphonodes PCR in 28 days (G1: 18% G2 67%). IFAT remission in 28 days (G1: 53%).	Study design.
**Ruiz G. et al. (2015)** [21]	Case report	N = 1	AM: 100 mg/kg/SID SC (3–6 weeks) A: 15 mg/kg/BID PO (long term)	Complete clinical remission in 28 days (90%).	Study design. No parasitological data (28–30 days).
**Solano-Gallego L. et al. (2016)** [18]	Case series	N = 37 dogs	AM: 80–100 mg/kg/SID SC (30 days) A: 10 mg/kg/BID (12 months)	Partial clinical remission in 30 days. Clinical-pathological remission in 28 days: Hematocrit increase (8.3%). Total protein decrease (2.5%). Globulin decrease (44%). IFAT remission in 28 days (44%). Parasitic remission blood PCR in 28 days (11.27%).	Study design.
**Gizzarelli M. et al. (2018)** [22]	Case report	N = 1	M: 2 mg/kg/SID PO (28 days) A: 10 mg/kg/BID PO	Complete clinical remission in 2 months.	Study design. No parasitological data (28–30 days).
**König M.L. et al. (2018)** [23]	Case report	N = 1	M: 2 mg/kg/SID PO (28 days) A at 10 mg/kg/BID (6 months)	Partial clinical remission in 30 days.	Study design. No parasitological data (28–30 days).
**Santos M.F. et al. (2019)** [7]	Case/Control Study	N (G1): 6/17 dogs N (G2): 6/17 dogs N (G3): 5/17 Healthy dogs (control group)	G1: AM: 100 mg/kg/SID SC (30 days) A: 10 mg/kg/BID PO G2: M: 2 mg/Kg/SID PO (30 days) A: 10 mg/kg/BID PO	Partial clinical remission in 30 days. Clinical-pathological in 30 days: G1: Hematocrit increase (8.62%). Total protein decrease (9%). Albumin increase (16%). Globulin decrease (33.6%). G2: Hematocrit decrease (6.34%). Total protein decrease (19.6%). Albumin increase (32%). Globulin increase (2.7%).	Study design.
**Dias Á.F.L.R. et al. (2020)** [12]	Cohort Study	N (G1): 15/45 dogs N (G2): 15/45 dogs N (G3): 15/45 dogs	G1: M: 2 mg/kg/SID PO (28 days) G2: M: 2 mg/Kg/SID PO (28 days) A: 10 mg/kg/BID PO (28 days) G3: A: 10 mg/kg/BID PO (28 days)	Partial clinical remission in 30 days (35%).	No clinical-pathological efficacy data (28–30 days). No parasitological data (28–30 days).
**Ayres E.D.C.B.S. et al. (2022)** [13]	Cohort Study	N (G1): 15/45 dogs N (G2): 15/45 dogs N (G3): 15/45 dogs	Group 1: M: 2 mg/kg/SID PO (29 days) Group 2: M: 2 mg/Kg/SID PO (29 days) A: 10 mg/kg/BID PO (29 days) Group 3: A: 10 mg/kg/BID PO (29 days)	Partial clinical remission in 29 days (58.4%). Clinical-pathological remission in 29 days: Hematocrit increase (18%). Total protein decrease (0.28%). Parasitic remission PCR skin in 29 days (95%).	No parasitological data (28–30 days).
**Gizzareli M. et al. (2023)** [17]	Case series	N = 173 dogs	M: 2 mg/kg/SID PO (28 days) A: 10 mg/kg/BID PO (2–12 months)	Complete or partial clinical remission 2.6 months (+/− 1.6). Clinical-pathological decrease 4.1 months (+/− 10.0). ELISA decrease 3.0 months (+/− 4.9).	Study design. No parasitological data (28–30 days).

## Data Availability

Data are contained within the article.

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
