# Peer review of "Critically Appraised Topic on Canine Leishmaniosis: Does Treatment with Antimonials and Allopurinol Have the Same Clinical and Clinicopathological Efficacy as Treatment with Miltefosine and Allopurinol, after One Month of Treatment?"

_vetsci, 2024, doi:10.3390/vetsci11060231_

Round 1
Reviewer 1 Report
Comments and Suggestions for Authors
The manuscript "Critically Appraised Topic on canine leishmaniasis: does treatment with antimonials and allopurinol have the same clinical and clinicopathological efficacy as treatment with miltefosine and allopurinol, after one month of treatment? " compares the efficacy of standard treatments (meglumine antimonials + allopurinol and miltefosine + allopurinol) in controlling clinical and clinicopathological changes associated with leishmaniasis after 28-30 days of treatment using critically appraised topic (CAT) methodology. The comparison is mainly based on aspefici parameters, while specific direct (PCR) and indirect (IFAT according to manual WOAH) diagnostic methodologies are not specified. The results showed that meglumine antimoniate combined with allopurinol showed greater clinical efficacy after one month of treatment than the combination of miltefosine and allopurinol, due to a significantly higher percentage of dogs achieving complete clinical remission within the first month of treatment with meglumine antimoniate with allopurinol. These results are already known in the outpatient clinic, so The manuscript while well presented, lacks innovation and significant contributions to the scientific literature. As a result, I believe this paper is not suitable for publication.
Author Response
The authors sincerely acknowledge and understand the comments of Reviewer 1. We are aware that this is a review article (following the structure and methodology of a Critical Appraisal of a Topic - CAT) and therefore does not provide new scientific information. However, we believe that, as you indicate, this is a topic of great interest due to the increasing prevalence and severity of canine leishmaniosis worldwide. Our manuscript analyzes and synthesizes the available information in a new way, until now not available in the scientific literature. For this reason, we think that it is a valuable contribution, that will be of interest to numerous veterinary clinicians and also researchers, and we think that it is worth publishing. In this article, the PCR and serological methods are not detailed because, as explained in detail in the discussion, in our review, the availability of data on the results of PCR techniques or serologic tests such as ELISA and IFAT performed 30 days after treatment was remarkably limited in most of the studies reviewed. This may be because in practice it is not common to perform these tests until 3 to 6 months after the start of treatment. One month after treatment the results of serology and PCR usually remain the same than at the day of the diagnosis.Reviewer 2 Report
Comments and Suggestions for Authors
The aim of this study was to compare the efficacy of standard treatments (meglumine antimoniate + allopurinol and miltefosine + allopurinol) in controlling clinical and clinicopathological alterations associated with leishmaniosis after 28-30 days of treatment using the critically appraised topic (CAT) methodology. From the study emerges the difficulty in comparing the two groups of publications using the two therapeutic combinations. The comparison is mainly based on nonspecific parameters, while specific direct and indirect methods are not significant or not specified.
this CAT and comparative analysis were to enhance our understanding of the potential distinctions among conventional leishmaniosis treatment strategies. The focus was on evaluating their clinical and clinicopathological efficacy, along with assessing the impact on parasite load and antibody levels after one month of treatment. The results showed that meglumine antimoniate combined with allopurinol exhibited greater clinical effectiveness after one month of treatment compared to the combination of miltefosine and allopurinol, due to a significantly higher proportion of dogs achieving complete clinical remission within the first month of treatment with meglumine antimoniate with allopurinol. The presented work, although addressing a relevant topic, lacks substantial innovation and significant contributions to scientific literature. The findings do not offer new perspectives or insights into the field of study and appear to reiterate or rehash concepts already widely explored. Consequently, I believe this paper is not suitable for publication in an academic-level scientific journal.
Author Response
The authors sincerely acknowledge and understand the comments of Reviewer 2. We are aware that this article is framed within a review that follows the structure and methodology of a Critical Appraisal of a Topic (CAT) and therefore does not present new scientific information. However, we believe that by analyzing and synthesizing the available information in a novel way, it adds value that has not previously been available in the scientific literature. For this reason, we consider it to be a significant contribution with the potential to be of interest to a wide range of veterinary clinicians and researchers. In this article, the direct and indirect methods are not detailed because, as explained in detail in the discussion, they are not relevant parameters for the 30 days of treatment administration. In our review, the availability of data on the results of serologic tests such as ELISA and IFAT performed 30 days after treatment was remarkably limited in most of the studies reviewed. This may be because in practice it is not common to perform these tests until 3 to 6 months after the start of treatment. One month after treatment the results of serology and PCR usually remain the same than at the day of the diagnosis.Reviewer 3 Report
Comments and Suggestions for Authors
[Veterinary Sciences] Manuscript ID: vetsci-2948274
Manuscript Revision
The manuscript is interesting and well-structured. Summary tables and graph are very nicely done and summarize important information, helping the readers to better understand studies and derived results. Searching across three databases (PubMed, CAB Abstracts, and Web of Science) was performed over several years to understand the clinical and clinical pathological effectiveness of two treatment options after one month of treatment (meglumine antimoniate and allopurinol or miltefosine and allopurinol). More information on scoring system used to define a clinical remission and more details for the clinical pathological evaluation needs to be provided. This can be done as written specifications or as a table. Limitations of the study are described with the discussion section. The paper is therefore accepted for publication with revisions.
Several points needs to be revised:
Material and method
Clinical scenario pag 2:
- “ELISA for anti- Leishmania infantum antibodies yielded a highly positive result”. Please specify in parenthesis how much was the titer and how you define a ”high antibodies level”.
Search strategy:
- Analysis strategy are clearly explained
Critical analysis:
- “To evaluate clinical efficacy, individuals who achieved a clinical score reduction of more than 90% were classified as having complete remission (CR), while those with a clinical score reduction of less than 90% were classified as having partial re-mission (PR). “ Please explain on which parameters the clinical score is based. Is this only resolution of the skin lesion? Please provide specifications. You can add this as written explanation or providing a table with the clinical parameters used and the scoring system provided
- “Clinical-pathological efficacy was evaluated by calculating the average percentage reduction of hematocrit, total serum proteins, albumin, and globulins.” Could you explain why albumin concentration should decrease? Why the decrease in hematocrit should correlate with an efficiency in the treatment? Please further explained the proposed inclusion parameters and their meaning.
- Since in the discussion section you wrote “The prevalent clinical-pathological manifestations observed in canine leishmaniosis encompass mild to moderate normocytic normochromic and non-regenerative anemia”, clinical-pathological efficacy should lead to an increase of the hematocrit. Please further comment on this.
Results
3.1. Meglumine antimoniate plus allopurinol:
- Globulin levels were assessed in three out of 16 articles evaluating 96 dogs. A decrease of 24%, 33.6%, and 44% was described in 54, six and 37 dogs, respectively. Please be consistent throughout the text. Use always values written as numbers.
- “Leishmania serology performed one month after treatment was evaluated only in two out of 16 studies”. Please see above comment.
- Please see above comment throughout the manuscript
Discussion
- “The prevalent clinical-pathological manifestations observed in canine leishmaniosis encompass mild to moderate normocytic normochromic and non-regenerative anemia, lymphopenia, hyperproteinemia characterized by elevated gamma-globulins and beta-globulins, along with hypoalbuminemia leading to a reduced albumin/globulin ratio and proteinuria”. As mentioned above please explain better why in material and method you showed a different explanation to the clinical-pathological efficacy.
- “It is suggested that dogs, particularly those with elevated antibody levels at diagnosis, may not exhibit a discernible reduction in antibody levels within the initial 30 days. Typically, a substantial decrease in levels becomes noticeable around 6 months, primarily influenced by the extended half-life of IgG”. So there is no association among the antibody titer level and the clinical remission? Please provide specification on what/how much is an “elevated antibody levels” at diagnosis. So the antibodies level at presentation are not important to the aim of remission?

Comments on the Quality of English LanguageMinor editing of English language required
Author Response
The authors greatly appreciate the comments from Reviewer 3, as well as her/his thorough review of the manuscript. We have carefully addressed all the suggested changes.
1. Clinical scenario:
“ELISA for anti- Leishmania infantum antibodies yielded a highly positive result”. Please specify in parenthesis how much was the titer and how you define a” high antibodies level”.
The concentration of Leishmania infantum antibodies in sera (375 EU) has been specified in parentheses. A high antibody level is defined as >300 EU (lines 59-60).
2. Critical analysis:
To evaluate clinical efficacy, individuals who achieved a clinical score reduction of more than 90% were classified as having complete remission (CR), while those with a clinical score reduction of less than 90% were classified as having partial re-mission (PR). “Please explain on which parameters the clinical score is based. Is this only resolution of the skin lesion? Please provide specifications. You can add this as written explanation or providing a table with the clinical parameters used and the scoring system provide.
Unfortunately, a unique clinical score was not used in all the studies analyzed. However, they evaluated several parameters such as dermatological lesions, weight loss, ocular lesions, fever, and others. To unify the analysis, we defined that individuals who achieved a clinical score reduction greater than 90% were classified as being in complete remission (CR), while those with a clinical score reduction of less than 90% were classified as being in partial remission (PR). To assess clinical efficacy, we compared the number of dogs achieving CR in both groups.
Clinical-pathological efficacy was evaluated by calculating the average percentage reduction of hematocrit, total serum proteins, albumin, and globulins.” Could you explain why albumin concentration should decrease? Why the decrease in hematocrit should correlate with an efficiency in the treatment? Please further explained the proposed inclusion parameters and their meaning.
Since in the discussion section you wrote “The prevalent clinical-pathological manifestations observed in canine leishmaniosis encompass mild to moderate normocytic normochromic and non-regenerative anemia”, clinical-pathological efficacy should lead to an increase of the hematocrit. Please further comment on this.
Thank you very much for this comment, as it has helped us to identify an error in the writing. Clinical-pathologic efficacy was evaluated by calculating the mean percentage reduction in total serum proteins and globulins, and the mean percentage increase in hematocrit and albumin. This has been changed in the text (lines 84-87).
3. Results:
Globulin levels were assessed in three out of 16 articles evaluating 96 dogs. A decrease of 24%, 33.6%, and 44% was described in 54, six and 37 dogs, respectively. Please be consistent throughout the text. Use always values written as numbers.
Leishmania serology performed one month after treatment was evaluated only in two out of 16 studies”. Please see above comment.
Please see above comment throughout the manuscript.
The suggested changes have been made throughout the manuscript.
4. Discussion:
“The prevalent clinical-pathological manifestations observed in canine leishmaniosis encompass mild to moderate normocytic normochromic and non-regenerative anemia, lymphopenia, hyperproteinemia characterized by elevated gamma-globulins and beta-globulins, along with hypoalbuminemia leading to a reduced albumin/globulin ratio and proteinuria”. As mentioned above please explain better why in material and method you showed a different explanation to the clinical-pathological efficacy.
As mentioned in section 3, the error has been corrected.
“It is suggested that dogs, particularly those with elevated antibody levels at diagnosis, may not exhibit a discernible reduction in antibody levels within the initial 30 days. Typically, a substantial decrease in levels becomes noticeable around 6 months, primarily influenced by the extended half-life of IgG”. So there is no association among the antibody titer level and the clinical remission? Please provide specification on what/how much is an “elevated antibody levels” at diagnosis. So the antibodies level at presentation are not important to the aim of remission?
The authors thank the reviewer for the comment because it has led to a text modification to make this paragraph clearer. The use of serology as a useful parameter for treatment monitoring has been a matter of debate. In fact, it is considered not useful in the short term due to the difficulty in detecting any clear reduction before 6 months of treatment in commercial assays (not end-point tests). This is especially true in dogs with high antibody levels. Therefore, the authors have changed “elevated antibody levels” by “high antibody levels” and have modified the whole paragraph to better explain this situation and add a reference (lines 210-224).
Reviewer 4 Report
Comments and Suggestions for Authors
The aim of this manuscript was to critically apprise the literature relating to the treatment of leishmaniosis in dogs; in particular the drug regimen used. For this the authors have reviewed several publications on this topic based around two treatment approaches. In doing this the authors have clearly defined the sources used, the approach to selection/ deselection of papers and summarised the findings of those used for the work. They have also identified the difficulties of comparing the data mined from the two sets of papers - these mostly related to the methodological differences in papers around detection of parasites or host parasite response. All this is very well described, however my only criticism is of the lack of detail around the statistical analysis both in the approach and rational for this. It would also be useful to see the collated data that was used to generate the analysis.
Based on the authors analysis - section 3.3 the discussion and conclusions would seem to be highly appropriate for the work, however I do think it would be good to recommend that a minimum standards of analysis should be used in papers looking at clinical outcomes for anti-infective drugs.
Author Response
The authors thank the reviewer for the comment, however, all the analyzed data have been already incorporated into the article. Should it be required, we are prepared to furnish the analyzed data in a supplementary document.
Reviewer 5 Report
Comments and Suggestions for Authors
The study provides the readers new fundamental information on the treatment and management of canine leishmaniasis. The study is well designed and organized mostly appropriately; besides, the manuscript is better to be considered, including the English language in terms of comprehension and fluency.
Major comments:
・Without line numbers in the manuscript, it is difficult to provide line by line comments/suggestions.
・Be checked all the references cited in the list throughout, following the journal instructions.
In Table 1,
・“Study design”, is better to be explained a little bit more concretely.
・Article; Year; Authors would be enough just to cite Reference Nos.
Comments on the Quality of English Language
The study is well designed and organized mostly appropriately; besides, the manuscript is better to be considered, including the English language in terms of comprehension and fluency.
Author Response
The authors are very grateful to the Reviewer 5 for her/his comments.
Without line numbers in the manuscript, it is difficult to provide line by line comments/suggestions.
The line numbers have been added to the manuscript.
Be checked all the references cited in the list throughout, following the journal instructions.
In Table 1, “Study design”, is better to be explained a little bit more concretely. Article; Year; Authors would be enough just to cite Reference Nos.
The suggested changes have been made in Table 1.
Round 2
Reviewer 1 Report
Comments and Suggestions for Authors
the reasons expressed by the authors convinced me, and the changes made improved the manuscript